# Dual-Band Dielectric Resonator Antenna with Filtering Features for Microwave and Mm-Wave Applications

**DOI:** 10.3390/mi14061236

**Published:** 2023-06-12

**Authors:** Mohamed Sedigh Bizan, Hassan Naseri, Peyman Pourmohammadi, Noureddine Melouki, Amjad Iqbal, Tayeb A. Denidni

**Affiliations:** Centre-Energie Matériaux et Télécommunications, Institut National de la Recherche Scientifique, Montréal, QC H5A1K6, Canada; hassan.naseri.gheisanab@inrs.ca (H.N.); peyman.pourmohammadi@inrs.ca (P.P.); noureddine.melouki@inrs.ca (N.M.); amjad.iqbal@inrs.ca (A.I.); tayeb.denidni@inrs.ca (T.A.D.)

**Keywords:** mm-wave band, microwave band, 5G, DRA, dual-band antennas, filtering

## Abstract

This paper presents a new design for a dual-band double-cylinder dielectric resonator antenna (CDRA) capable of efficient operation in microwave and mm-wave frequencies for 5G applications. The novelty of this design lies in the antenna’s capability to suppress harmonics and higher-order modes, resulting in a significant improvement in antenna performance. Additionally, both resonators are made of dielectric materials with different relative permittivities. The design procedure involves the utilization of a larger cylinder-shaped dielectric resonator (D_1_), which is fed by a vertically mounted copper microstrip securely attached to its outer surface. An air gap is created at the bottom of (D_1_), and a smaller CDRA (D_2_) is inserted inside this gap, with its exit facilitated by a coupling aperture slot etched on the ground plane. Furthermore, a low-pass filter (LPF) is added to the feeding line of D1 to eliminate undesirable harmonics in the mm-wave band. The larger CDRA (D_1_) with a relative permittivity of 6 resonates at 2.4 GHz, achieving a realized gain of 6.7 dBi. On the other hand, the smaller CDRA (D_2_) with a relative permittivity of 12 resonates at a frequency of 28 GHz, reaching a realized gain of 15.2 dBi. The dimensions of each dielectric resonator can be independently manipulated to control the two frequency bands. The antenna exhibits excellent isolation between its ports, with scattering parameters (S_12_) and (S_21_) falling below −72/−46 dBi at the microwave and mm-wave frequencies, respectively, and not exceeding −35 dBi for the entire frequency band. The experimental results of the proposed antenna’s prototype closely align with the simulated results, validating the design’s effectiveness. Overall, this antenna design is well-suited for 5G applications, offering the advantages of dual-band operation, harmonic suppression, frequency band versatility, and high isolation between ports.

## 1. Introduction

During the past few decades, researchers have become increasingly interested in DRA antennas due to their distinct benefits. The DRA, or dielectric resonator antenna, offers a range of benefits such as compact size, the ability to operate over a broad range of frequencies, and straightforward excitation [1,2]. Another advantage is that DRAs are made entirely of dielectric material, meaning their loss can be minimal even at frequencies in the mm-wave band. Combining these characteristics makes the DRA ideal for microwave and mm-wave applications [3,4,5,6]. In terms of dual-band DRAs, several designs have been suggested. For instance, Xiao-Chuan Wang presented a design that utilizes a cross-slot-coupled dualband (RDRA) [7]. This design realized a right-hand circular polarization. The antenna exhibits an impedance bandwidth of 11.41% and 8.41%, 3-dB axial ratio bandwidths of 2.11% and 2.21%, and antenna gains greater than 5.41 and 4.31 dBi, respectively. Several dual-band or wideband DRA antennas have been proposed in recent studies. In [8], an RDRA was presented with a frequency range of 3.41 to −3.59 GHz and 5.11 to 5.9 GHz. In [9], a hybrid dual-band antenna was proposed with the following two operation frequencies: 2.4 GHz and 5.8 GHz. Another wideband dual-feed DRA was presented in [10], which utilized a slot with an L-shaped and a vertical copper micro-strip to excite the antenna. In [11], a dual-band/wideband quasi-Yagi antenna is proposed, which utilizes the (TE_11_) and (TE_13_) modes of the DRA to provide two close-operation frequencies. The experimental impedance bandwidth is 21.71% with a peak realized gain of 8 dBi. Additionally, an array antenna consisting of many elements of CDR antenna resonating at 28 GHz was proposed in [12] by Niayesh and Kouki. The antenna discussed in [12] has a bandwidth of 9.82% at 28.73 GHz, achieving an efficiency of 89% and realized gain of 15.69 dBi. Other works have focused on developing dual-band filtering dielectric resonator antennas, such as the one presented in [13], which was tested at 7.71 and 13.36 GHz with impedance bandwidths of 11.41% and 4.81% and maximum realized gains of 7.66 and 10.51 dBi, respectively. In [14], a hybrid antenna that combines three resonators was proposed to create four operation frequency bands. A compact shared-aperture antenna with a significant frequency ratio is proposed in [15]. The dual-band implementation excites a high-frequency dielectric resonator antenna (DRA) and a low-frequency slot antenna. The operating frequencies of these two antenna parts can be tuned independently. The authors in [16] combine a liquid Dielectric Resonator Antenna (DRA) and a patch configuration to achieve multi-band and multi-mode operation for Wi-Fi applications in the 2.4 GHz and 5 GHz frequency bands. The proposed approach addresses the issue of undesirable higher-order modes in the DRA by suppressing and replacing them with desired modes through a hybrid design. The dielectric resonator (DR) in the antenna design in [17] exhibits the excitation of four resonant modes. This design achieves dual passbands with distinct frequencies and bandwidths in each channel. Additionally, the antenna benefits from two pairs of orthogonal degenerate modes and the chosen excitation principle, ensuring high isolation between the two channels that reach -38 dB. The authors in [18] presented a dual-band antenna design that combines a millimeter-wave substrate-integrated dielectric resonator antenna (SIDRA) beam-steerable array with a long-term evolution (LTE) folded monopole antenna (FMA). The SIDRA array is integrated within the FMA’s clearance area, allowing for a shared substrate and aperture. The LTE FMA operates at 1.78–2.62 GHz (38.2%) with a peak gain of 3.9 dBi, while the MMW array covers 26.4–29.8 GHz (12.2%) with a peak gain of 10 dBi and supports ±45° beam-steering angles. Ref. [19] presents a hybrid antenna design that combines strip, slot, and dielectric resonator antenna (DRA) resonators, generating four resonances within the frequency bands of 28 and 39 GHz. The lower frequency band of 26.41–30.42 GHz is covered using strip and slot modes, while the upper-frequency band of 36.05–40.88 GHz is covered by employing the (TE_111_) and (TE_131_) modes of the DRA. In a previous study [20], the authors presented an innovative design that utilizes a dual dielectric resonator antenna (DRA) with opposite orientations. This unique antenna design offers dual-band filtering characteristics and a quasi-isotropic radiation pattern. The upper cylindrical ceramic of the antenna is fed by a conformal strip and a circular patch, while the lower one is fed by a circular aperture. This feeding mechanism generates (HEM_11δ_) and (HEM_12δ_) modes within the ceramic material at frequencies of 2.8 and 5.4 GHz, respectively, resulting in a dual-band response. However, none of the references mentioned have introduced a dual-band DRA that resonates in both sub-6 GHz and millimeter-wave bands, which is essential for 5G applications where wireless systems need to operate in both microwave and mm-wave bands. Therefore, there is interest in developing this type of DRA.

The authors in [21] propose a two parallel-plate waveguide resonator antenna for microwaves and the Fabry–Perot resonator antenna for millimeter-waves, all combined into one structure. The double-fed dual-band frequency antenna utilizes an air gap in the DRA [22,23]. In [24], the authors introduce a windowed slow-wave parallel-plate waveguide (WSW-PPW). By incorporating electromagnetic bandgap (EBG) structures into a dual-layer printed PPW, the WSW-PPW enables the transmission and integration of MW antennas while blocking MMW propagation. However, these structures lack filtering capabilities to eliminate unwanted harmonics in the mm-wave band and do not provide independent control of the frequency bands. In addition, they use hybrid resonators, such as the dielectric resonator and Fabry–Perot resonator or patch antenna, to operate in two bands, which limits their ability to control the frequency bands independently. Therefore, there is a need for a dual-band DRA that operates at both microwave and mm-wave frequencies with high isolation levels, independently of frequency bands, and has the ability to suppress harmonics.

This paper introduces a dielectric resonator dual-band antenna for 5G applications operating in microwave and millimeter-wave frequency ranges. The proposed design employs two cylindrical dielectric resonators (CDRs) that are nested together and excited by vertical conducting strips and a coupling aperture slot. The larger CDR resonates at 2.4 GHz with a realized gain of 6.7 dBi, while the smaller CDR resonates at 28 GHz and has a realized gain of 15.2 dBi. Most importantly, LPF suppresses any unwanted harmonics in the mm-wave band from the microwave band. Furthermore, independence of frequency bands are achieved, which helps control each band without any effect on the other. Thus, with all this in mind, it is asserted that the suggested dual-band structure introduces some new features to dual-band DRAs and makes them more suitable for 5G systems supporting both sub-6 GHz and millimeter-wave bands. Moreover, the antenna provides a high isolation level. (S_12_) and (S_21_) fall to (−72/−46 dBi) at the desired frequencies and do not reach above (−35 dBi) for all bands.

## 2. Antenna Design

This section describes the steps taken to reach the final design of a new dual-band, double-cylinder DRA antenna. The proposed antenna configuration and its design evolution are shown in Figure 1 and Figure 2.

In the first step, as in Figure 2a, a large dielectric resonator is designed for the microwave band to operate at 2.4 GHz. The CDRA comprises Hik500 (E r = 6 and loss tangent = 0.002). This material has been chosen due to its availability in the customer market for a reasonable price. The radius (*a*) and height (*h*) of a CDRA can be obtained using the following equation [4]:(1)$$f\left(\mathrm{GHz}\right)=\frac{30k_{o}a}{2h(\frac{a}{h})}$$
where (*a/h*) is the aspect ratio of the CDR, and (*k*_o_) is the free space wave number. The value (*k*_o_*a*) can be calculated for the (TE_011_) mode as follows:(2)$$k_{o}a=\frac{1}{\sqrt{r+1}}\left(1+0.7013\left(\frac{a}{h}\right)-0.002713{\left(\frac{a}{h}\right)}^{2}\right)$$

Equations (1) and (2) provide the initial estimates of the values, but they are not precise because they do not consider how the feed network affects the DRA. This can cause changes in the center frequency and limitations in bandwidth. In addition, an infinite ground plane assumption underlies the equations.

After establishing the large microwave band resonator (D_1_), a hollow region is created on the bottom to provide room for the small mm-wave band resonator (D_2_). By doing so, the frequency at which resonance occurs increases. To maintain the frequency at 2.4 GHz, optimization is performed on the dimensions of (D_1_). Figure 3 illustrates the impact of the hollow on the (S_11_) of (D_1_).

Then, a small dielectric resonator (D_2_) is inserted in the air gap for the mm-wave band. The (D_2_) is designed to resonate at 28 GHz by following the same steps as (D_1_). (D_2_) is made of Hik500, with E r = 12 and loss tangent = 0.002. Finally, to suppress the harmonics that (D_1_) causes in the millimeter-wave band, an LPF is added to the feeding transmission line of (D_1_) to eliminate unwanted harmonics, especially those occurring at 28 GHz. By comparing Figure 4 and Figure 5, we can clearly observe the filter’s impact. Adding this filter to the proposed antenna not only raises the level of isolation between the two ports but also improves the reflection coefficient of both the mm-wave band (S_11_) as well as microwave band (S_22_). As it is clearly noted that (S_12_) and (S_21_) drop down to (−72/−46 dBi) at the desired frequencies and not above (−35 dBi) for the whole bands.

(D_1_) and (D_2_) are mounted and stacked into two layers of RO3003 material substrate with different thicknesses (1.52 mm for the bottom and 0.13 mm for the top). A cut is made in the middle of the bottom thick substrate, reaching 5 mm beyond its center. A rectangle mutual coupling slot is etched on the ground plane at a 0.7 mm length and 2.8 mm width to excite the fundamental mode of (D_2_), while the (TE_011_) mode is excited in (D_1_) by a vertical copper tap glued on it. The only purpose of stacking these two substrate layers is to provide a thin substrate for the mm-wave feeding microstrip line and a thick substrate for the microwave transmission line. Figure 6 illustrates the combined components of the proposed antenna.

The optimized parameters of the proposed antenna, according to Figure 2a step 2, are shown in Table 1.

## 3. Parameteric Study

A parametric study was carried out using CST software to adjust the dimensions of CDR1 and CDR2. To match impedance, the feed line was lengthened by 1mm beyond the center of CDR2. The reflection coefficient and realized gain were examined for various values of dr_1_, dr_2_, and dh_2_. Figure 7. shows that we can shift the realized gain of the mm-wave band by adjusting the height of (D_2_). Furthermore, the operating frequency of the microwave band (2.4 GHz) can be tuned by monopolizing the radius of the big resonator dr_2_; meanwhile, there is no impact on the millimeter-wave band resonance, as illustrated in Figure 8.

In the same way, varying D1’s radius dr_1_ causes a frequency shift in mm-wave with the stability in microwave resonant frequency of (D_1_) (see Figure 9). Figure 8 and Figure 9 prove the independence of the two bands from each other. In this structure design, the ability to control the resonance frequency of every band separately is realized.

## 4. Experimental Results

In order to confirm the accuracy of the optimized simulated results, a model of the proposed antenna was created and physically constructed, as depicted in Figure 10. The Agilent ENA series vector network analyzer was utilized to determine the antenna’s reflection coefficient. The prototypes of two embedded cylindrical resonators were made of Hik500 material with a different permittivity of 6 and 12, respectively. Both resonators were laid on the two-substrates layer of Roger RO3003 with thicknesses of 0.13 mm and 1.52 mm, respectively. To excite it, a copper microstrip was glued on the big (D_1_).

Figure 11 illustrates the experimental and simulated reflection coefficients and the realized gain of both bands (millimeter-wave and microwave bands) with a reasonable agreement. There is a small shift in frequency towards the right side, likely caused by various experimental factors, such as mistakes made during machining and the potential presence of small spaces between the DRAs and the ground plane. The figure illustrates that the antenna gains were measured and simulated at frequencies in the mm-wave bands of 15.2 and 14.6 dBi at 28 and 28.2 GHz, respectively. The measured antenna gain is slightly lower than the simulated gain of 0.7 dB. On the other hand, for the microwave band, the measured reflection coefficient remains almost the same as the simulated one (just a 20 MHz frequency shift to the right side), while the realized gain decreases by 0.5 dBi from 6.7 dBi to 6.2 at 2.4 GHz.

The proposed design’s simulated and measured radiation patterns are plotted in Figure 12. A good level of agreement between the simulated and measured results can be observed. The cross-polarization fields are weaker than their co-polar peers by more than −30 dB. Cross-polarization isolation is essential for 5G antenna design. It helps reject interference, maintain signal integrity, enable frequency band versatility, achieve isolation between ports, and enhance system performance and signal quality. By effectively isolating signals of different polarizations, the antenna can operate efficiently and deliver reliable performance in its intended 5G applications.

## 5. Discussion

The proposed antenna design is developed, simulated, and experimentally evaluated at 2.4 and 28 GHz for both the mm-wave and microwave bands. This concept has significant advantages, such as the proposed antenna having one dual-band DRA-based structure resonating at both microwave and millimeter-wave bands. Additionally, it has a high isolation level as (S_12_) and (S_21_) are less than (−35 dBi) for all bands. In addition, the antenna has a harmonics suppression capability provided by the LPF. The antenna is independently tunable. Each band of the dual-band double CRDA antenna is independently controllable. The lower resonant frequency can be controlled by varying the dimensions of CDR1. The higher resonant band can be controlled by modifying the dimensions of the CDR2.

Table 2 contains a comparison of the dual-band antennas proposed in the work. In [21,22,23,24], the authors designed antennas based on hybrid structures, including a Fabry–Perot, patch antenna and DRA to cover mm-wave 24/28 GHz and microwave 2.4/3.5 GHz bands; however, they fail to eliminate the harmonics that are generated due to the microwave resonator affecting the mm-wave band. Furthermore, they do not mention if their designs have frequency-band independence. Conversely, due to the LPF, the proposed work has the advantages of not including any unwanted harmonics at the mm-wave band and achieving the total frequency independence of bands. Furthermore, both resonators are dielectric resonators.

## 6. Conclusions

This paper presents the design and fabrication of a cylindrical dielectric resonator antenna that operates on two different frequency bands for 5G applications. The proposed design integrates two cylindrical dielectric resonators (CDRs) with different permittivities of 6 and 12, which are embedded and excited to operate in two frequency bands: sub-6 GHz and millimeter-wave bands. The large CDR resonates at 2.4 GHz and has a 6.7 dBi realized gain, and the fundamental mode is excited using a copper tape glued on its circumference, which is connected to a microstrip line. In contrast, the small CDR (D_2_) operates at 28 GHz, reaching a 15.2 dBi realized gain and the fundamental mode is excited using a rectangular coupling slot. To eliminate unwanted harmonics at the mm-wave band, a low-pass filter (LPF) was added to the (D_1_) feeding line. This resulted in an improvement in the isolation level and reflection coefficients of both frequency bands. The antenna exhibits independence of frequency bands, making it a unique feature of this design. The simulated and experimental results show good agreement, indicating the effectiveness of the design. In summary, the novel characteristics introduced in the design of the dual-band microwave and mm-wave DRA make this antenna a promising candidate for 5G applications. 

## Figures and Tables

**Figure 1 micromachines-14-01236-f001:**
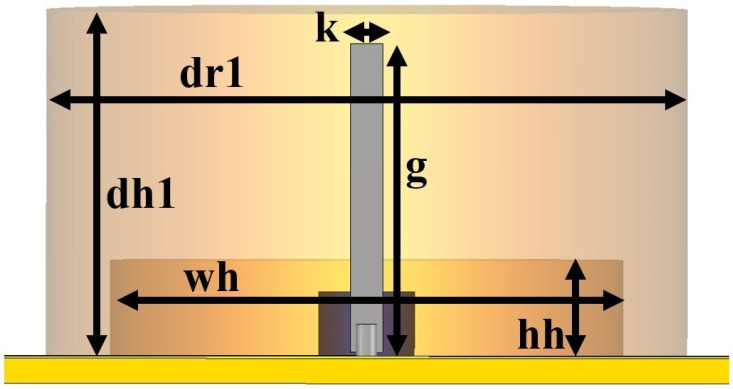
(D_1_) dimensions.

**Figure 2 micromachines-14-01236-f002:**
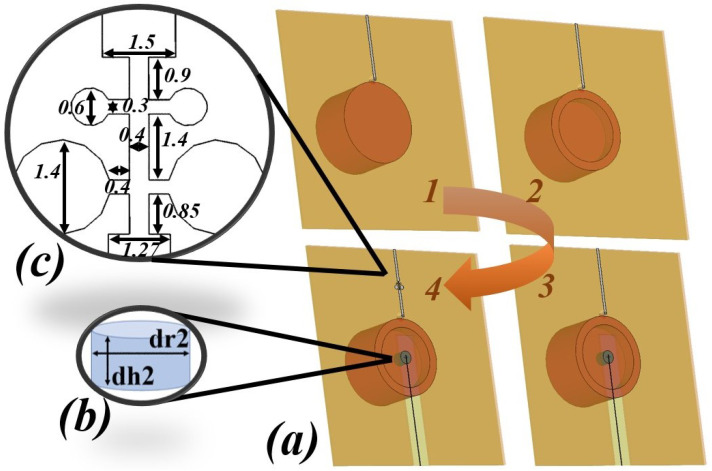
The proposed antenna’s geometrical structure, (**a**) the four steps of the proposal antenna, (**b**) (D_2_) dimensions, (**c**) LPF configuration.

**Figure 3 micromachines-14-01236-f003:**
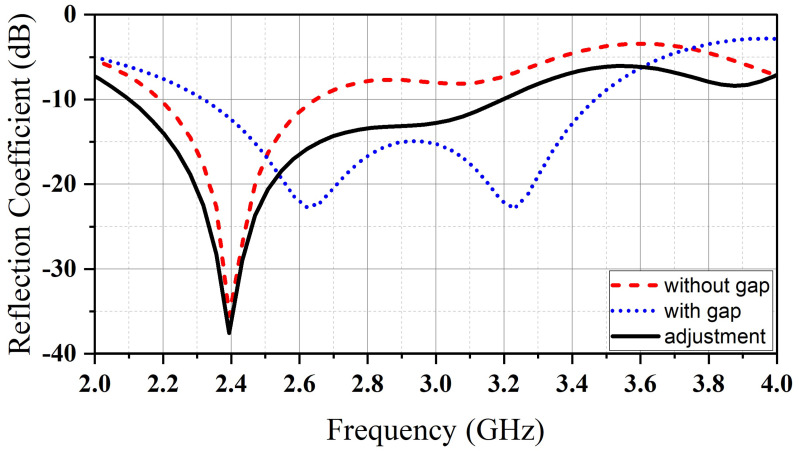
The simulated reflection coefficient associated with (D_1_).

**Figure 4 micromachines-14-01236-f004:**
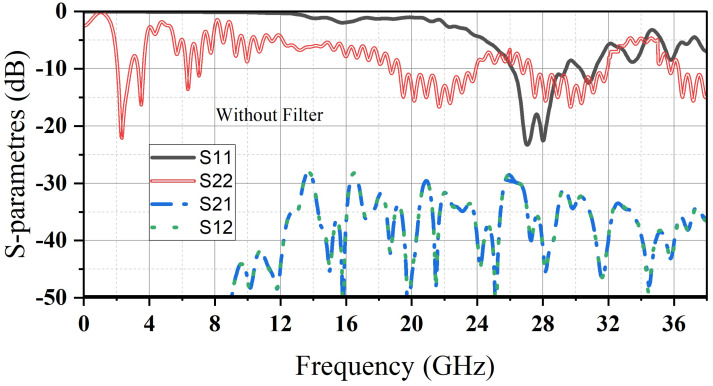
S-parameters of the proposed antenna before adding the LPF.

**Figure 5 micromachines-14-01236-f005:**
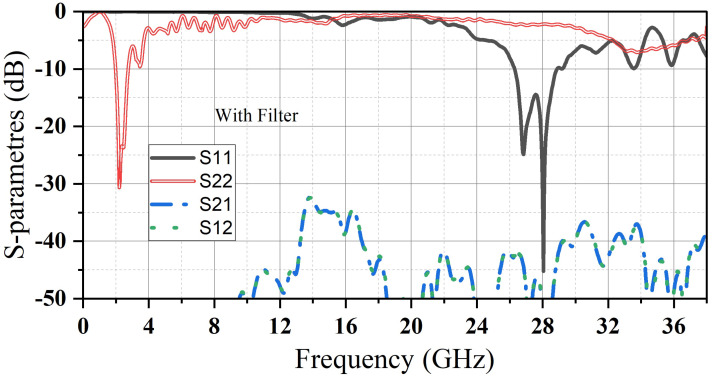
S-parameters of the proposed antenna after adding the LPF.

**Figure 6 micromachines-14-01236-f006:**
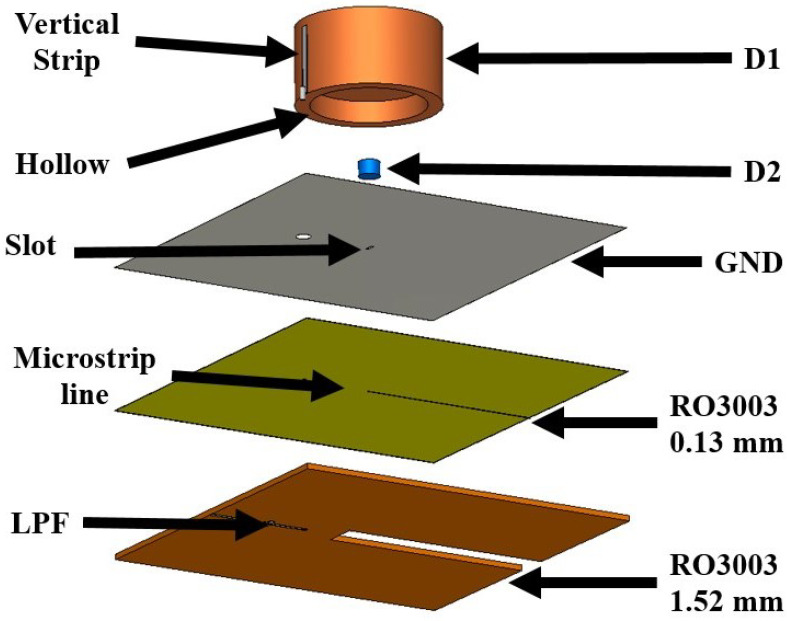
The proposed antenna’s layers.

**Figure 7 micromachines-14-01236-f007:**
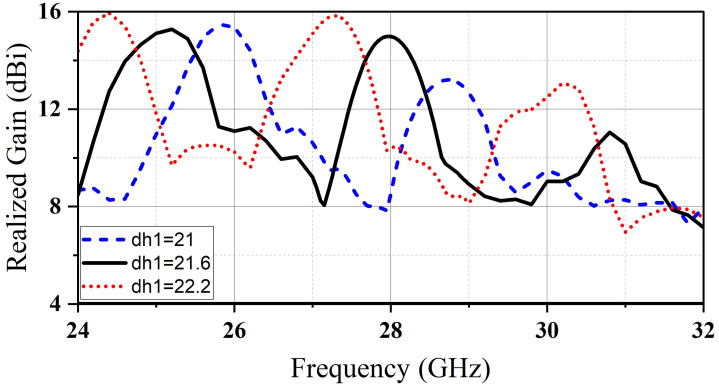
The realized gain of the mm-wave band for various values of dh_1_.

**Figure 8 micromachines-14-01236-f008:**
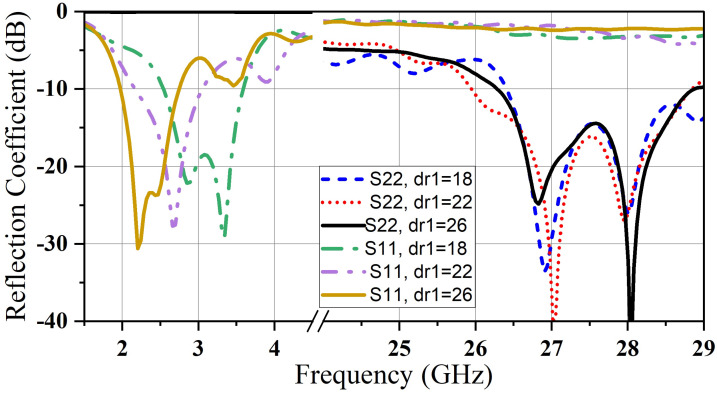
Reflection coefficient for various values of dr_1_.

**Figure 9 micromachines-14-01236-f009:**
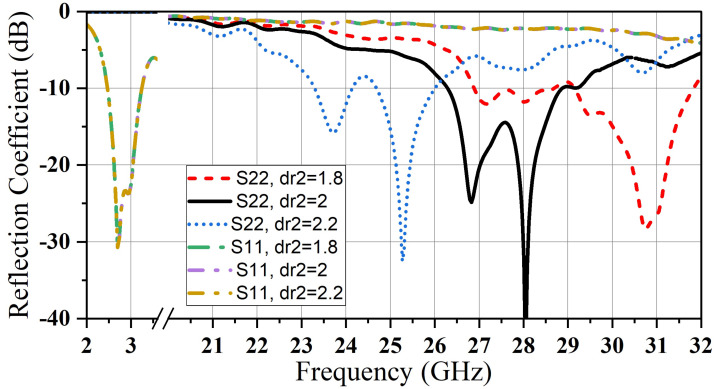
Reflection coefficient for various values of dr_2_.

**Figure 10 micromachines-14-01236-f010:**
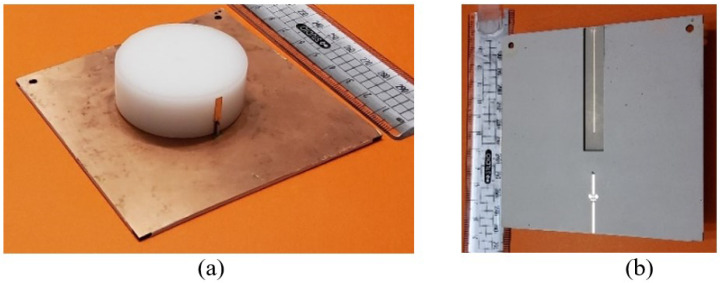
The proposed antenna prototype (**a**) top view and (**b**) bottom view.

**Figure 11 micromachines-14-01236-f011:**
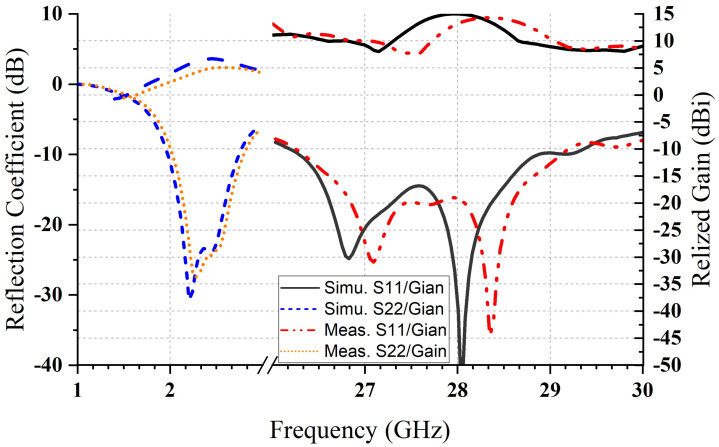
Reflection coefficient and realized gain, measured, and simulated.

**Figure 12 micromachines-14-01236-f012:**
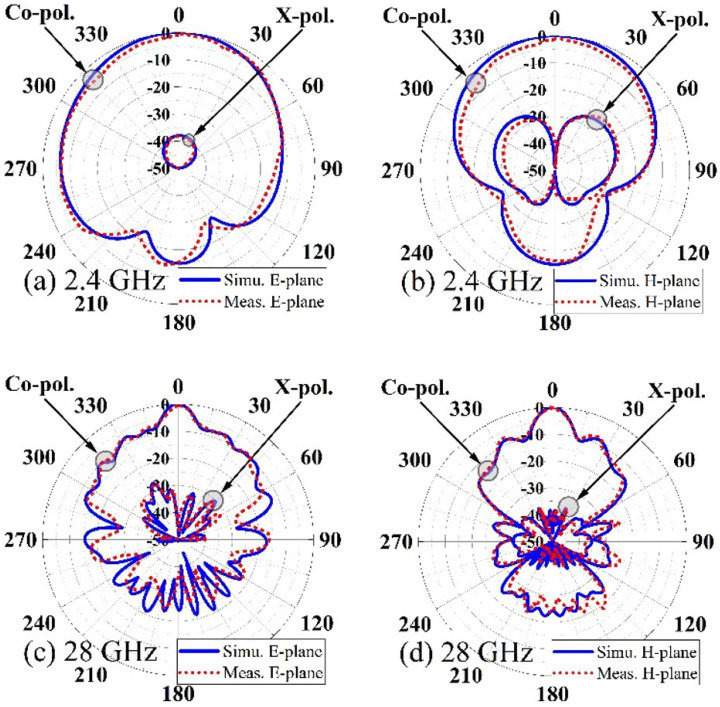
Simulated and measured radiation patterns, (**a**) E-plane at 2.4 GHz, (**b**) H-plane at 2.4 GHz, (**c**) E-plane at 28 GHz, (**d**) H-plane at 28 GHz.

**Table 1 micromachines-14-01236-t001:** The Proposed Antenna’s Dimensions in Millimeters.

Parameter	*w*	*dr* _1_	*dr* _2_	*wh*	*k*
Value (mm)	100	26	2.0	15	2.0
Parameter	*L*	*dh* _1_	*dh* _2_	*hh*	*g*
Value (mm)	100	21.6	3.2	6	19

**Table 2 micromachines-14-01236-t002:** Comparison of this work and other work.

Parameter	[21]	[22]	[23]	[24]	This Work
Resonator type (microwave/mm-wave)	WRA /FPRA	DRA /FPRA	DRA /FPRA	Patch /DRA	DRA /DRA
Resonance Frequency (GHz)	2.4/24	2.4/24	2.4/24	3.5/28	2.4/28
Realized Gain (dBi) (microwave/mm-wave)	7.23/11.26	6.81/18.2	6.71/11.93	4.95/13.94	6.7/15.2
Isolation Level (dB) (microwave/mm-wave)	N/A	N/A	N/A	−36/−40	−72/−46
Harmonic Suppression Capability	No	No	No	No	Yes
Frequency Response Independently	N/A	N/A	N/A	N/A	Yes

## Data Availability

Not applicable.

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
