# Peer review of "Dual-Band Dielectric Resonator Antenna with Filtering Features for Microwave and Mm-Wave Applications"

_micromachines, 2023, doi:10.3390/mi14061236_

Round 1

Reviewer 1 Report

Dual-band Dielectric Resonator Antenna with Filtering Features for Microwave and Mm-wave Applications has been proposed. The following are observations/suggestions / possible modifications:

 1.     It is recommended that the abstract should contain the majority of the statistical details of the antenna. The abstract should ideally have a line of novelty proposed in the antenna.

2.     The introduction needs to be strengthened. Please ensure to have the latest research included in the literature review.

3.     The figure 1 inset figure lacks visibility. It can be shown separately.

4.     Page 2 last line. Only tan is mentioned please complete it if a loss tangent needs to be written. You may also incorporate a line showing the Hik500 availability in the consumer market – Easy / Hard to avail.

5.     The adjustment spell is incorrect in Figure 2. Please also write the reflection coefficient instead of the shorter form.

6.     The X-axis and Y-axis text is going beyond the figures. Please put them adequately.

7.     The text within the figure needs improvement across the manuscript.

8.     The importance of cross-pol isolation should be shown appropriately.

9.     Is it possible to add another parametric analysis to show the antenna's sensitivity to its mechanical dimensions?

10.  Conclusion is adequately mentioned.

11.  The references can be significantly improved. The recent literature would add value to the literature review in the manuscript.

Please ensure to do a grammar check of the manuscript.

Reviewer 2 Report

This is an interesting idea and as the paper shows a dual band capability with 2.4GHz and 28GHz. It is 5G in part but 2.4GHz is WiFi so it is more mmWave/WiFi that it is offering.

It would be interesting to see the fields for the TE modes at the two frequencies as is common analysis for DRAs. That is the only thing of technical value I can see worth adding.

I note though some minor points to note.

There are a number of notation errors:
Units like GHz should not be italic.
Variables a and h need to be italic.
The 0 in K_0 should be subscriot and not italic.
"tan = 0.02" has a delta character missing. This is also a problem elsewhere too
"r = 6" has epsilon missing. Also epsilon missing in other places.
Table 1 needs a caption and needs to have the variables showing properly as notations.
Table 2 also needs a caption and tidying up.

Do also check for typographical errors with a proof check to find errors such as the following:
"FurtherMORE, independence of frequency bands ARE achieved,..."
"..a hollow REGION is made on.."
'adjustment' not 'adjastement' in Fig. 2

I have noted before some typographical errors to correct.

Round 2

Reviewer 1 Report

The suggested modifications have been carried out.